# Are cardiovascular health measures heritable across three generations of families in Soweto, South Africa? A cross-sectional analysis using the random family method

Lisa J Ware [ID],[1,2] Innocent Maposa,[3] Andrea Kolkenbeck-Ruh,[1,4]
Shane A Norris [ID],[1,5] Larske Soepnel,[1,6] Simone Crouch,[1] Juliana Kagura,[3]
Sanushka Naidoo,[1] Wayne Smith,[7] Justine Davies[8,9]

For numbered affiliations see end of article.

**Correspondence to**
Dr Lisa J Ware;
lisa.ware@wits.ac.za

## ABSTRACT

**Objectives** Cardiovascular disease is increasing in many low and middle-income countries, including those in Africa. To inform strategies for the prevention of cardiovascular disease in South Africa, we sought to determine the broad heritability of phenotypic markers of cardiovascular risk across three generations.

**Design** A cross-sectional study conducted in a longitudinal family cohort.

**Setting** Research unit within a tertiary hospital in a historically disadvantaged, large urban township of South Africa.

**Participants** 195 individuals from 65 biological families with all three generations including third-generation children aged 4–10 years were recruited from the longest running intergenerational cohort study in Africa, the Birth to Twenty Plus cohort. All adults (grandparents and parents) were female while children were male or female.

**Primary and secondary outcome measures** The primary outcome was heritability of blood pressure (BP; brachial and central pressures). Secondary outcomes were heritability of arterial stiffness (pulse wave velocity), carotid intima media thickness (cIMT) and left ventricular mass indexed to body surface area (LVMI).

**Results** While no significant intergenerational relationships of BP or arterial stiffness were found, there were significant relationships in LVMI across all three generations (p<0.04), and in cIMT between grandparents and parents (p=0.0166). Heritability, the proportion of phenotypic trait variation attributable to genetics, was estimated from three common statistical methods and ranged from 23% to 44% for cIMT and from 21% to 39% for LVMI.

**Conclusions** Structural indicators of vascular health, which are strong markers of future clinical cardiovascular outcomes, transmit between generations within African families. Identification of these markers in parents may be useful to trigger assessments of preventable risk factors for cardiovascular disease in offspring.

## STRENGTHS AND LIMITATIONS OF THIS STUDY

⇒ Intergenerational transmission was evaluated for a range of indicators of cardiovascular health within urban African families.

⇒ The sample included biological family members from three generations.

⇒ Heritability estimates were compared for three commonly used statistical methods.

⇒ The sample size is a limitation with the random family statistical method used to increase the numbers of comparisons available.

⇒ Only maternal family members were included.

## INTRODUCTION

Within South Africa, a quarter of all adults are hypertensive and one in five deaths are from cardiovascular disease (CVD).[1] CVD mortality and morbidity are set to rise with increasing life expectancy (now at 64 years; an increase of 10% in the last decade)[2] and increasing levels of overweight and obesity (68% women, 31% men).[3] Much focus is placed on detecting and treating CVD, but with limited healthcare resources, pragmatic approaches are needed including primary prevention in younger, at-risk individuals to prevent CVD.[4]

Estimation of heritability or the proportion of variation in a phenotypic trait between individuals that is attributable to genetic factors has been used for many years to predict disease risk in medicine.[5] While there may be debate regarding the exact measurement of genetic, environmental and interaction effects on trait variability, broadly heritability indicates the degree of resemblance of a trait within biological families.[6] There is evidence that strong predictors of future adverse

cardiovascular (CV) outcomes (such as heart attacks and strokes) may be transmitted through biological families so that measures in parents or grandparents may identify children at future risk.[7]

Early vascular predictors of CVD outcomes include both structural (eg, thickening or stiffening of arterial walls, cardiac hypertrophy) and functional changes (eg, elevated blood pressure (BP)).[8–12] Hypertension is the largest contributor to CVD in Africa, with research showing elevated BP in children as young as 5 years of age.[13] Studies of monozygotic and dizygotic twins have shown high heritability of systolic blood pressure (SBP) and diastolic blood pressure (DBP) in populations of both African and European decent,[14 15] though heritability may be lower for individuals of African decent.[16] Within South Africa, data are also emerging that BP is heritable across families (parent–child and sibling–sibling pairs).[17] However, due to the high levels of hypertension in South African adults, hypertension in a family member is unlikely on its own to be a sensitive enough indicator to identify at-risk young adults or children for intervention.

As such, additional measures may be needed to identify those family members most at risk and where early intervention may have greater returns. Evidence from outside of Africa has shown that several other markers of CVD risk are heritable. For example, central BP may show stronger heritability than the brachial BP typically measured in routine care.[18] Also, carotid artery structure, function and pathology have been shown as heritable, with diameter and carotid intima media thickness (cIMT) appearing as the most heritable traits.[19–21] Furthermore, arterial stiffness, as assessed by pulse wave velocity (PWV), has also been reported as heritable within family studies,[18 22] and findings from echocardiography studies suggest that several cardiac measurement parameters may be heritable within families, including left ventricular (LV) function and structure including LV mass (LVM) and LV hypertrophy (LVH).[23–26] Indeed, the combination of arterial stiffness and central pressure has been suggested as a potential tool to investigate risk in nuclear families.[27]

However, there is limited evidence from African families to indicate which indicators of CV health are most related and, therefore, potentially most useful to indicate intergenerational risk within family units in South Africa. One previous study suggested that echocardiography may be particularly useful to detect intergenerational transmission of changes in cardiac structure and function in South African families (parent–child and sibling–sibling pairs),[28 29] though how this and other vascular measures are related across children, parents and grandparents in the region is not known. Additionally, the frequent background of undernutrition and burden of infectious diseases may mean that heritability estimates in Africa are different from elsewhere.

Therefore, we sought to investigate how a range of indicators of CV health (brachial and central pressures, arterial stiffness, cIMT and echocardiography findings) were related within three generations (grandparents, parents and children) of African families from urban South Africa to inform further risk identification and potential targeted CVD prevention efforts.

## METHODS
### Study population and sample size
Biological families with three generations (grandmother, mother and child (boy or girl aged 4–10 years)) were recruited from the largest and longest running birth cohort study in Africa: the Birth to Twenty Plus (Bt20-plus) cohort described in detail previously.[30 31] Families in this cohort are tracked over time through engagement in ongoing assessments. In 2019, a database of 162 Bt20-plus index children (now the mothers) was drawn from all previous Birth to Twenty assessments that indicated both survival of their biological mother and birth of a biological child. These index children were then contacted by telephone to confirm the presence of their biological mother, and a biological child between the ages of 4 and 9 years. Families with participants who were pregnant, experiencing current acute illness or with any major congenital disorders were excluded. All eligible families were invited to take part. The study design was a cross-sectional in-depth assessment of vascular health at a research unit located within the grounds but operating independently of the outpatient and inpatient services of a large tertiary government hospital in Soweto, a historically disadvantaged township in South Africa. Data were collected between August 2019 and March 2020. Previous work in East African families found high heritability of BP (systolic, diastolic and pulse pressure $h^2$ of 0.37, 0.24 and 0.54), though the authors did not assess other vascular measures.[32] Based on these previous reported levels of heritability between two generations and using the methods of Klein,[33] n=65 families (n=195 individuals) at alpha=0.05 would give 82% power to detect an $h^2$ of 0.4 and 94% power to detect an $h^2$ of 0.5 in BP. With three generations, these estimates may be conservative.

### Ethical considerations
Trained researchers who spoke the participant's home language explained the study and all participants provided written informed consent prior to taking part in the study. For children, the mother of the child provided written consent, with children aged 7 years and above also giving their written assent to take part. We used the Strengthening the Reporting of Observational Studies in Epidemiology cohort checklist when writing our report.[34]

### Patient and public involvement
The study design was informed by previous work with two generations from this cohort, where participants expressed a desire to include additional generations in CV health assessments. However, participants were not involved in the study design, recruitment or conduct of the study. During 2023, a series of workshops are planned with the community to disseminate results and

to explore the cocreation of potential community-level interventions.

## Measurements

Standard protocols were used for collection of all data, with the same staff repeating all measures or assessments of interoperator variability conducted as described further in the online supplemental appendix 1. Medical history (including antihypertensive medication use) and health behaviours were recorded via self-report. Tobacco use (daily or occasional current use of both smoked and smokeless tobacco products) was assessed using questions from the Global Adult and Tobacco Survey.[35] Alcohol use was evaluated using the WHO Alcohol Use Disorders Identification Test (WHO-AUDIT),[36] with hazardous or harmful alcohol use assessed as an AUDIT-C score (first three questions—shortened form) of ≥3 and/or a total AUDIT score of ≥8.

Trained researchers measured height and weight in triplicate to the nearest 0.1 cm and 0.1 kg using a portable stadiometer and electronic scale (SECA, Hamburg, Germany). Waist and mid-upper arm circumference were measured in triplicate to the nearest 0.1 cm following standard measurement protocols.[37 38]

All measures were taken in the morning following an overnight fast and with no caffeine or tobacco for at least 3 hours prior to measurement. Using the SphygmoCor Excel device (AtCor Medical, Naperville, USA) with appropriate size brachial cuff, brachial BP and resting heart rate were determined, and central arterial pressures (central SBP, central DBP, pulse and mean arterial pressure) were estimated. Three measurements were taken, with the second and third measures averaged for analysis. Ultrasound measures were taken in triplicate with the Mindray DC-70 Ultrasound System (Mindray, Shenzhen China). Further detail for these assessments is provided in the online supplemental appendix 1.

## Analyses

The primary outcome was heritability of BP (brachial and central pressures). Secondary outcomes were heritability of arterial stiffness (PWV), cIMT and left ventricular mass indexed to body surface area (LVMI). All exposure effects were adjusted for age, height, weight and sex in the regression models, with heritability estimates adjusted for age.

For adults, body mass index (BMI; $kg/height\ m^2$) was categorised as follows: <18.5 underweight; 18.5–24.9 normal weight; 25.0–29.9 overweight; ≥30 obese. Children's BMI was categorised as underweight, normal, overweight or obese using age-specific and sex-specific cut-offs from the International Obesity Task Force.[39] Waist to height ratio was calculated for both adults and children, as this has previously been shown as a predictor of health risks of obesity across the life course in all ethnic groups.[40] In adults, prehypertension was defined as 120–139 mm Hg systolic or 80–89 mm Hg diastolic and not currently taking antihypertensive medication, while

hypertension was defined as BP ≥140 mm Hg systolic or ≥90 mm Hg diastolic or currently taking antihypertensive medication. For children, elevated BP was defined using the age, sex and height-adjusted percentiles of the American Academy of Pediatrics Clinical Practice Guideline (2017).[41]

The Devereux formula was used to calculate LVM[42] and left ventricular mass index (LVMI) was calculated as a ratio of LVM indexed to body surface area (BSA).[43] LVH was defined as LVMI >95 $g/m^2$ for adult women and LVMI >95th percentile for children. Normality of data was checked with visual inspection of histograms and the Shapiro-Wilk test.[44]

Our analyses followed two stages: stage 1 determining the association between parent–offspring pairs for each of the vascular health traits, and stage 2 estimating heritability for traits that exhibited an association in the parent–offspring pairs. Participant characteristics and the associated vascular health measurements are also described.

### Stage 1. Random family method

In this study, we used the random family method as described in detail by Usuzaki et al[45] and implemented the analysis based on Heß[46] randomisation inference algorithm. We used resampling of the exposure variable to generate the distribution of parental trait effect on offspring, controlling for confounding variables as below. We used the classical model generally used to explore heritability in phenotypic traits:

$$y_i = \beta_0 + \tau z_i + \beta X + \epsilon_i,$$

where $y_i$ is the offspring trait, and $\tau$ is the 'treatment' effect (regression slope) for $z_i$, the parental trait. $X$ is a matrix of control variables and $\beta$ the associated coefficients. $\tau$ is obtained for the original pairs $(z_i, y_i)$ and using randomisation inference tests, we performed 5000 resampling-based pairs to obtain the distribution of the statistic $\tau$, that is, the distribution of random parental trait effect on offspring's corresponding trait. Randomisation inference tests have the advantage that they can handle small sample sizes and do not rely on validity of the specified model regardless of the generated statistic being from the model.[46] Randomisation inference also produces the distribution of a test statistic under a designated null hypothesis, thereby allowing us to assess whether the observed (original parent–offspring pair) relationship statistic (regression coefficient) is significantly different and hence the null hypothesis can be rejected in favour of the parental trait having a significant influence on the offspring trait. In brief, regression coefficients were generated for all primary and secondary CV measures within the biological families: adjusting brachial and central pressures, PWV and cIMT for age, height, weight and sex; and adjusting LVMI for age and sex only as it is already indexed to BSA. Restricted resampling of the data was then employed to generate 5000 random family units ensuring random pairing of parent–offspring biological

families. The regression coefficients for each CV outcome marker were then compared between the family pair and random pairs. Kernel density plots of $\tau$ values for original family pairs and random pair $\tau$ values were then generated to assess statistical significance of the selected traits.

## Stage 2. Heritability estimation

For those variables which showed significantly greater association between family members compared with randomly generated pairs using the random family method, heritability estimate(s) were derived using the variance components decomposition method based on the linear mixed-effects model (LMM) as all vascular health traits of interest were continuous. The restricted maximum likelihood (ReML) method was used to estimate the variance components and hence heritability. However, due to concerns by Hadfield *et al*[47] and Morrissey *et al*[48] on ReML limitations we additionally implemented the Bayesian method for variance components and heritability estimation,[49] thereby creating a range for each heritability estimate. The basic model (LMM) is:

$$Y|Z, X \sim N\left(X\beta, G\sigma_g^2 + I_n\sigma_e^2\right),$$

where additive genetic variance of the trait $G$ is estimated using relatedness information between individuals or genotype $Z$ with both fixed-effects $\beta$ for $X$ control variables, $\epsilon_i \sim N\left(0, \sigma_e^2\right)$, and random effects following a normal distribution with mean 0 and variance $G\sigma_g^2$.[50] $G$ is the genetic relatedness matrix (GRM) and was estimated using the kinship package in R (V.4.0.2).[51] We also used the kinship package to plot the pedigree of one family in our data set. The Bayesian linear mixed model with polygenic effects ($g$) has the following sampling model:

$$y|\beta, u, \sigma^2 \sim N\left(X\beta + Zu, \sigma^2 I\right), \beta \sim N\left(0, \sigma_\beta^2 B\right), u \sim N\left(0, \sigma^2 G\right),$$

where $B$ is known and non-singular diagonal matrix and $\sigma_\beta^2$ as a hyperparameter was used. The $G$ in $\sigma^2 G$ is the GRM estimated through the kinship package for the family relatedness. Note that for this model the likelihood and assumed priors were:

$$y_i \sim N\left(\mu, \sigma^2 I\right)$$
$$\mu = X\beta + g$$
$$\beta_j \sim N\left(0, 1000^2\right), \forall j = 1, \ldots, p$$
$$g \sim N\left(0, \sigma_g^2 G\right)$$
$$\sigma_g^2 \sim InvGamma\left(s_1, s_2\right)$$
$$\sigma^2 \sim InvGamma\left(s_1, s_2\right),$$

where $s_1$ and $s_2$ are chosen to provide non-informative priors. We used interface software with R (rJAGS[52] and rSTAN[53]) to perform Markov chain Monte Carlo (MCMC) and Hamiltonian Monte Carlo (HMC) simulations, respectively.[50] Heritability was then computed as $h^2 = \frac{\sigma_g^2}{\sigma_g^2 + \sigma^2}$. The marginal distributions of all parameters and estimation of the best linear unbiased predictions

for the model were obtained using Gibbs' sampling (MCMC) and the leap-frog integration method (HMC). The samplers made 100 000 simulations and only results of the last 90 000 were used in the inference. We used two Bayesian paradigms to enable comparisons and manage the inherent uncertainty associated with estimating genetic variance components[47] as well as in using small sample sizes. Age of the participant was used as a control variable for all models and was standardised together with the vascular health traits before estimation to improve efficiency of Bayesian sampling.

## RESULTS

Of the 162 index children identified, n=48 (30%) could not be contacted as either the telephone number had changed or they did not respond to calls or voice messages; n=14 (9%) did not wish to take part; n=5 (3%) were not eligible due to current illness, pregnancy or a biological child not in the required age range; n=4 (2%) were no longer residing in Soweto; n=3 (2%) were not available due to school or work commitments; and n=9 (6%) booked appointments but did not attend. Finally, 65 families (49% of those contacted) took part in the study providing n=130 adults and n=65 children and generating 195 biological pairings: 130 first generation and 65 second generation.

Whole family completion rates for the vascular measures were as follows: carotid ultrasound (n=63); brachial BP, heart rate and pulse wave analysis (n=62); echocardiography (n=59); PWV (n=40); and all vascular measures (n=40). Families with complete anthropometry data and at least one vascular measurement complete for a family pairing (parent/child, grandparent/parent or grandparent/grandchild) were included in the analysis as the random family method does not require all three generations to have data, only that a family has one or more biological pairs with valid measurements. Descriptive characteristics are presented in table 1, including the number of adults and children with successful measurements for each variable.

Median age of grandparents, parents and children was 56, 29 and 7 years, respectively. All parents and grandparents were female while 45% of children were male. Among adults, 92% of grandparents and 77% of parents were overweight or obese. While the majority of children were a healthy weight (65%), one in five were overweight or obese. Elevated BP (prehypertension or hypertension) was present in 88% of grandparents, 46% of parents and 27% of children. In general, markers of CVD risk worsened with age (table 1), with 5% of children, 29% of parents and 45% of grandparents categorised as having an LVH.

## Results of random family and heritability analysis

Table 2 shows the results from comparing biological family pairs to randomly generated non-biological pairings, with statistically significant associations observed

**Table 1** Characteristics of the n=65 included families (grandparents, parents and children)

| | Grandparents n=65 | Parents n=65 | Children n=65 |
|---|---|---|---|
| Age (years) | 56 (10) | 29 (0) | 7 (3) |
| Female, n (%) | 65 (100) | 65 (100) | 36 (55) |
| **Anthropometry** | | | |
| Height (cm) | 157.3 (8.1) | 159.5 (7.5) | 122.5 (16.2) |
| Weight (kg) | 83.4 (25.9) | 72.4 (22.7) | 23.8 (9.3) |
| Mid-upper arm circumference (cm) | 36.3 (7.4) | 32.8 (8.7) | 18.3 (4.4) |
| Waist circumference (cm) | 104.4 (18.2) | 88.1 (21.8) | 54.8 (12.2) |
| Waist to height ratio | 0.67 (0.12) | 0.57 (0.15) | 0.44 (0.07) |
| Body mass index (BMI; kg/m$^2$) | 34.5 (10.6) | 29.3 (9.3) | 15.7 (2.2) |
| Underweight, n (%) | 1 (2) | 1 (2) | 9 (14) |
| Normal weight, n (%) | 4 (6) | 14 (21) | 42 (65) |
| Overweight, n (%) | 12 (18) | 19 (29) | 12 (19) |
| Obese, n (%) | 48 (74) | 31 (48) | 2 (3) |
| **Medical history and health behaviour** | | | |
| Previous diabetes diagnosis, n (%) | 4 (6) | 0 | – |
| Previous hypertension diagnosis, n (%) | 41 (63) | 4 (6) | – |
| On antihypertensive medication, n (%) | 40 (62) | 2 (3) | – |
| Currently uses tobacco, n (%) | 18 (28) | 11 (17) | – |
| Harmful/hazardous alcohol use, n (%) | 10 (15) | 22 (34) | – |
| **SphygmoCor: pulse wave analysis** | n=65 | n=65 | n=62 |
| Brachial measures | | | |
| Systolic blood pressure (SBP; mm Hg) | 133 (28) | 117 (18) | 103 (11) |
| Diastolic blood pressure (DBP; mm Hg) | 80 (16) | 73 (12) | 63 (9) |
| Resting heart rate (bpm) | 65 (15) | 69 (12) | 80 (14) |
| Blood pressure (BP) status, n (%) | | | |
| Normal/healthy BP | 8 (12) | 35 (54) | 45 (73) |
| Elevated BP/prehypertension | 13 (20) | 22 (34) | 5 (8) |
| Hypertension | 45 (68) | 8 (12) | 12 (19) |
| Central measures (c) | | | |
| cSBP (mm Hg) | 126 (26) | 106 (16) | 92 (12) |
| cDBP (mm Hg) | 81 (16) | 74 (11) | 64 (8) |
| Pulse pressure (mm Hg) | 42 (14) | 33 (8) | 28 (4) |
| Mean arterial pressure (mm Hg) | 99 (19) | 87 (15) | 79 (12) |
| **SphygmoCor: pulse wave velocity** | n=57 | n=61 | n=56 |
| Carotid-femoral PWV (m/s) | 8.45 (1.83) | 6.50 (0.88) | 4.33 (0.64) |
| **Ultrasound carotid measurements** | n=63 | n=63 | n=63 |
| Carotid IMT (cIMT; left side, mm) | 0.66 (0.18) | 0.50 (0.10) | 0.44 (0.09) |
| **Ultrasound cardiac measurements** | n=58 | n=63 | n=63 |
| LVM indexed to body surface area (LVMI_BSA, g/m$^2$) | 91.4 (36.4) | 82.8 (36.4) | 56.4 (21.5) |
| Left ventricular hypertrophy, n (%) | 26 (45) | 18 (29) | 3 (5) |

Data are presented as median (IQR) unless otherwise indicated. For children, LVH was defined as LVMI >95th percentile (109.4 g/m$^2$).
LVH, left ventricular hypertrophy; LVM, left ventricular mass; LVMI, left ventricular mass index; PWV, pulse wave velocity.

within families for cIMT between grandparents and parents, and for LVMI between all first-degree generations. Combining the heritability estimates from the different methods (table 3) showed that heritability of cIMT ranged from 0.234 to 0.439 such that between 23% and 44% of the variation in cIMT was explained by heritability within families. For LVMI, the estimates from the various methods were closer, suggesting between 21% and 39% of the variation in LVMI was explained by heritability within families. Importantly, though the heritability

**Table 2** Results of random family analysis

| Outcome | Exposure | Observed effect* [T(obs)] | c | n | P value=c/n |
|---|---|---|---|---|---|
| Brachial SBP—GC | Brachial SBP—GP | 0.029 | 3123 | 5000 | 0.625 |
| Brachial SBP—GC | Brachial SBP—P | 0.123 | 1027 | 5000 | 0.205 |
| Brachial SBP—P | Brachial SBP—GP | 0.109 | 967 | 5000 | 0.193 |
| Brachial DBP—GC | Brachial DBP—GP | −0.006 | 4647 | 5000 | 0.929 |
| Brachial DBP—GC | Brachial DBP—P | 0.063 | 2676 | 5000 | 0.535 |
| Brachial DBP—P | Brachial DBP—GP | 0.001 | 4970 | 5000 | 0.994 |
| Central SBP—GC | Central SBP—GP | −0.005 | 4649 | 5000 | 0.930 |
| Central SBP—GC | Central SBP—P | 0.075 | 2249 | 5000 | 0.450 |
| Central SBP—P | Central SBP—GP | 0.094 | 1392 | 5000 | 0.278 |
| Central DBP—GC | Central DBP—GP | 0.028 | 3379 | 5000 | 0.676 |
| Central DBP—GC | Central DBP—P | 0.119 | 1180 | 5000 | 0.236 |
| Central DBP—P | Central DBP—GP | 0.006 | 4702 | 5000 | 0.940 |
| PWV—GC | PWV—GP | −0.006 | 4655 | 5000 | 0.931 |
| PWV—GC | PWV—P | 0.166 | 766 | 5000 | 0.153 |
| PWV—P | PWV—GP | 0.104 | 1038 | 5000 | 0.208 |
| cIMT—GC | cIMT—GP | 0.093 | 962 | 5000 | 0.192 |
| cIMT—GC | cIMT—P | 0.171 | 1445 | 5000 | 0.289 |
| cIMT—P | cIMT—GP | 0.133 | 83 | 5000 | **0.017** |
| LVMI_BSA—GC | LVMI_BSA—GP | −0.076 | 2301 | 5000 | 0.460 |
| LVMI_BSA—GC | LVMI_BSA—P | 0.242 | 213 | 5000 | **0.043** |
| LVMI_BSA—P | LVMI_BSA—GP | 0.277 | 102 | 5000 | **0.020** |

Variables with p<0.05 are shown in bold.
*All exposure effects were adjusted for age, height, weight and sex in the regression models. P is the empirical probability value; c is the number of absolute effects ≥ the observed targeted generation effect (eg, grandparent on grandchild, grandparent on parent, etc, as indicated by the formula below); n is the number of generated pseudorandom families assessed on the targeted generation effect to determine c, where c=#{|T|≥|T(obs)|}.
cIMT, carotid intima media thickness; DBP, diastolic blood pressure; GC, grandchild; GP, grandparent; LVMI_BSA, left ventricular mass indexed to body surface area; P, parent; PWV, pulse wave velocity; SBP, systolic blood pressure.

estimates from the different estimation methods were related (online supplemental figure 1) and each parameter overlapped, high SD for phylogenetic variance estimates as well as heritability estimates was observed.

## DISCUSSION

The aim of this study was to examine a range of phenotypic markers of CV risk across three generations to determine the degree to which these measures of vascular health are transmitted through generations in an urban South African family cohort, and give an indication of whether these findings in older generations can be used to trigger assessments of CV risk in younger generations. While we did not find significant heritability of BP, possibly due to the high prevalence of elevated BP and hypertension across all generations, our results do suggest that, in this population, structural markers of CV risk (intima media thickness (IMT) in the common carotid artery (CCA)

**Table 3** Heritability estimates from different methods

| | cIMT (mm) | | | LVMI_BSA (g/m$^2$) | | |
|---|---|---|---|---|---|---|
| | ReML | MCMC | HMC | ReML | MCMC | HMC |
| Phylogenetic variance (p) | 0.131 (0.114) | 0.310 (0.101) | 0.175 (0.111) | 0.180 (0.172) | 0.405 (0.141) | 0.240 (0.154) |
| Error variance | 0.426 (0.070) | 0.385 (0.056) | 0.416 (0.065) | 0.660 (0.107) | 0.603 (0.085) | 0.647 (0.095) |
| Phenotypic variance | 0.556 (0.080) | 0.695 (–) | 0.591 (–) | 0.840 (0.122) | 1.008 (–) | 0.887 (–) |
| Heritability ($h^2$) | 0.234 (0.179) | 0.439 (0.098) | 0.282 (0.146) | 0.214 (0.182) | 0.394 (0.099) | 0.258 (0.139) |
| β* | 0.709 (0.048) | 0.705 (0.048) | 0.708 (0.047) | 0.496 (0.059) | 0.496 (0.059) | 0.493 (0.060) |

*Coefficient for age which was adjusted for all models for both vascular markers.
cIMT, carotid intima media thickness; HMC, Hamiltonian Monte Carlo; LVMI_BSA, left ventricular mass indexed to body surface area; MCMC, Markov chain Monte Carlo; ReML, restricted maximum likelihood.

(cIMT) and LVM (LVMI)) are heritable across African generations. This supports the intergenerational transmission of CV risk and identifies potential markers for the detection of at-risk families.

To our knowledge, there is scant information to date on the degree to which these phenotypic markers of CV risk are heritable within African families. However, the heritability estimates we identified for these structural CV markers are similar to those reported in several previous studies from research outside of Africa. For example, our estimates for heritability of cIMT (23%–44%) are similar to the 38% heritability reported in 586 families from the Framingham Heart Study[21] and the 34% reported in Latino parent–offspring pairs (69 families).[54] However, our estimates are lower than the 56% heritability reported from 100 Dominican families in the Northern Manhattan Study[55] and slightly higher than the 21% estimate reported in 32 American Indian families from the Strong Heart Family Study.[19] Lower estimates may be related to the pedigrees included in the samples. For example, the Strong Heart Family Study included first, second, third, fourth and greater degree relatives, while the other studies included only first-degree relatives. Further studies in first-degree relatives from 76 families in France provide a similar cIMT heritability estimate of 30%.[56] Given our finding that significant heritability was observed in first-degree relatives (grandparent-parent), our results broadly agree with other studies and may be among the first to identify this heritability in families in Africa.

We also saw broad agreement between our heritability estimates for LVMI (21%–39%), with estimates from studies outside of Africa including the Framingham Heart Study (30% heritability between parent–child pairs),[24] from 52 white European families (23%) and from 368 Chinese families living in Taiwan (27%).[23 57] Again, our estimate is higher than that from the Strong Heart Study (17%)[58] and lower than that from the Northern Manhattan Study (49%).[59] Our estimates are also lower than those from 169 hypertensive Japanese families living in Hawaii (43%)[60] and from the HyperGEN study (46%; 527 families, 51% African American; 53% hypertensive).[61] Generally, these higher heritability estimates for LVMI are from studies including or exclusively involving hypertensive participants. However, this may not in itself explain the higher estimates as we included family members with hypertension, as did the GENOA study (Genetic Epidemiology Network of Arteriopathy) in African-American hypertensive siblings with 34% estimated heritability of LVMI,[26] falling within the range of our findings.

When comparing our LVMI heritability estimates with the one study found within Africa (from 181 nuclear families in our same urban township in South Africa),[28] our estimates are lower. However, this study indexed LVM to height rather than BSA, with other studies showing this produces higher indexed LVMI values.[62] Importantly, the agreement between the studies that LVMI is heritable within families in this region supports the need for improved screening services.

Our findings for BP were not expected and are contrary to other studies where BP heritability has been observed within families. In a systematic review and meta-analysis by Kolifarhood et al,[16] heritability of SBP and DBP was observed across regions ranging from 17% to 52% for SBP and from 19% to 41% for DBP, though estimates were lower in African populations. However, African data were scarce with one study in Nigeria from Adeyemo et al[63] reporting heritability estimates of 34% for SBP and 29% for DBP in 528 families including 1825 individuals. While this was a large sample, heritability of BP has been observed in smaller African studies. For example, Bochud et al[32] found a significant heritability estimate for office SBP of 28% in 314 East African (Seychellois) adults from 76 families. However, in this study, family members were recruited for having at least two siblings with hypertension and family relationships included first-degree (sibling pairs, parent–offspring pairs), second-degree (grandparent–grandchild pairs, avuncular pairs, ie, uncle/aunt–niece/nephew) and third-degree (first cousin pairs) relatives. Our research included only first-degree and second-degree relatives in whom heritability might be expected to be higher, though our overall sample size (n=198) was smaller.

We also expected to find significant heritability for arterial stiffness within our families. Data from the Framingham Heart Study (1480 individuals from 817 families) suggest around 40% heritability of carotid-femoral PWV.[22] While evidence from a study in Brazil (125 families, 1675 individuals) shows a lower heritability estimate (27%),[64] this study also included first, second and third-degree relatives. To our knowledge, our results may be some of the first to investigate the intergenerational heritability of carotid-femoral PWV as a measure of arterial stiffness in families within South Africa and, possibly, in Africa highlighting the need for further work in African families, perhaps increasing sample size through the inclusion of third-degree relatives.

Given constrained resources for CVD treatment in the region, pragmatic and targeted prevention approaches are needed leveraging measurements that may be taken as part of routine clinical practice. Given the heritability of the factors identified in this study, we are not suggesting that people should be screened for these factors to identify at-risk children and families. Rather that offspring of adults in whom these factors are found should be targeted for rigorous assessment of risk, especially for raised LVM where this is measured in clinical practice.

### Strengths and limitations

Our findings must be viewed in light of the limitations of this research, most notably the small sample size resulting in high SDs observed for phylogenetic variance estimates as well as heritability estimates. However,

our heritability estimates from the different estimation methods for each parameter overlap giving confidence for our analysis, and the heritability estimates observed for cIMT and LVMI are similar to many of those reported previously. Additionally, the number of families included in this analysis is similar or more than many other heritability studies, with the random family method increasing the numbers of comparisons available. While our findings contribute to the small but growing evidence base for Africa, further research is needed across the continent to assess the generalisability of our results.

A further limitation results from the individuals in which we could not collect all phenotypic markers of CV risk, most notably the SphygmoCor PWV and the echocardiography measures. This difficulty was in part due to excess body mass, for example, the mean adult BMI of those with unsuccessful echocardiography measurement was $40.9 \pm 10.5 \, \text{kg/m}^2$. We also did not collect data on family history or blood markers of CV risk such as cholesterol within this study. Future studies should consider inclusion of a full CVD risk panel. Our lack of 24-hour ambulatory blood pressure monitoring (ABPM) data within families is also a limitation and future studies should consider the use of ABPM where feasible, as heritability estimates appear higher for ABPM than for office BP.[65] While we have successfully used ABPM in South African adults previously,[66] this was significantly more challenging in this urban cohort with young children and our attempts were not successful. Community-based support for families during ABPM measurement may be helpful in the future.

While it is noted that comparison with other studies can be problematic due to different populations, methods, study designs and environmental influence on phenotypic variance as highlighted by North *et al*,[19] we have taken care to compare our results only to studies that are methodologically similar. For example, all comparisons for LVMI heritability presented here include only studies using echocardiographic measurement of LVM, as LVM heritability estimates from electrocardiography may be higher.[25] Furthermore, heritability estimates for IMT often vary between the CCA and the internal carotid artery, with heritability estimates frequently higher for CCA, so that it is important to compare results for IMT measured in the same location. Furthermore, it is noted that heritability estimates between and within populations are not constant and are influenced by factors such as environmental changes and migration.[5] While this may limit the generalisability of findings from any one study, it remains that heritability estimates for these CV phenotypes appear largely similar across many of the studies, regions and populations.

A key strength of this research is the contribution of evidence for the heritability and intergenerational transmission of CV health in black African families living in an urban African township, including children prior to adolescence, and the comparison of several different methods to estimate heritability. Further, the high levels of elevated BP and hypertension observed in our population across older and younger adults and in the children reinforce the need for prevention programmes early in life.

## CONCLUSION

Our results suggest that structural CV indices in the CCA and in the left ventricle of the heart are heritable within African families. Where adults are identified with elevated cIMT or LVH, screening should be conducted in first-degree and second-degree relatives, especially to identify younger individuals most at risk of later poor vascular health, where prevention efforts may yield the greatest returns. Better understanding of the factors that promote transmission of poor vascular health from one generation to the next will support development of interventions to break the upward spiral of CVD on the continent.

**Author affiliations**
[1]SAMRC Developmental Pathways for Health Research Unit, Faculty of Health Sciences, University of the Witwatersrand, Johannesburg, South Africa
[2]DSI-NRF Centre of Excellence in Human Development, University of the Witwatersrand, Johannesburg, South Africa
[3]Division of Epidemiology and Biostatistics, School of Public Health, University of the Witwatersrand Faculty of Health Sciences, Johannesburg, South Africa
[4]Cardiovascular Pathophysiology and Genomics Research Unit, School of Physiology, Faculty of Health Sciences, University of the Witwatersrand, Johannesburg, South Africa
[5]Global Health Research Institute, School of Human Development and Health and NIHR Southampton Biomedical Research Centre, University of Southampton, Southampton, UK
[6]Julius Global Health, Julius Center for Health Sciences and Primary Care, Utrecht, The Netherlands
[7]Hypertension in Africa Research Team (HART), MRC Research Unit for Hypertension and Cardiovascular Disease, North-West University, Potchefstroom, South Africa
[8]Institute of Applied Health Sciences, University of Birmingham, Birmingham, UK
[9]Wallenberg Research Centre at Stellenbosch University, Stellenbosch Institute for Advanced Study, Stellenbosch, South Africa

**Contributors** LJW, JD, SAN and IM conceived the idea for the manuscript and designed the analyses. IM and LJW performed the analyses. LJW, IM, JD, JK, SN, AK-R, LS, SC, WS and SAN all contributed to the interpretation of the results. All authors contributed to drafting the manuscript and have seen and approved the final version. LJW is the guarantor for this work and accepts full responsibility for the work.

**Funding** This work was supported by the Wellcome Trust (UK) (grant number: 214082/Z/18/Z).

**Competing interests** JD is a member of the Trial Steering Committee for D-Clare (UK MRC-funded study: MR/T023562/1) for which no payment is received. She is also a member of the DSMB for NIH-funded study (5R01HL144708) for which an honoraria of $200 is received. She has received the standard $400 NIH honoraria for being a panel member of their Implementation Science Grant funding stream and is a member of the WHO Working Group to discern targets for the Diabetes Compact.

**Patient consent for publication** Not applicable.

**Ethics approval** This study involves human participants and the Human Research Ethics Committee (Medical) of the University of the Witwatersrand approved the protocol (Ref: M190263). Participants gave informed consent to participate in the study before taking part.

**Provenance and peer review** Not commissioned; externally peer reviewed.

**Data availability statement** Data are available upon reasonable request. Data are available upon request from SAN.

**ORCID iDs**
Lisa J Ware http://orcid.org/0000-0002-9762-4017
Shane A Norris http://orcid.org/0000-0001-7124-3788

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
