## [Reviewer comments · BMJ Open]

ARTICLE DETAILS

TITLE (PROVISIONAL)	Are cardiovascular health measures heritable across three generations of families in Soweto, South Africa? A cross-sectional analysis using the random family method.
AUTHORS	Ware, Lisa; Maposa, Innocent; Kolkenbeck-ruh, Andrea; Norris, Shane; Soepnel, Larske; Crouch, Simone; Juliana, Kagura; Naidoo, Sanushka; Smith, Wayne; Davies, Justine

VERSION 1 – REVIEW

REVIEWER	Maria Capra Guglielmo da Saliceto Hospital
REVIEW RETURNED	24-Feb-2022

GENERAL COMMENTS	The authors of this interesting article conduct a study to determine the heritability of phenotypic markers of cardiovascular risk across three generations. In their manuscript, they state that in South Africa “one in five deaths are from cardiovascular disease”, so cardiovascular disease (CVD) is an issue of utmost importance in this Country. I find this article interesting as it gives us a perspective of cardiovascular risk in three generations, even if the generations are only female, as it is stated in the study limitations. I think this article is written accurately. However, I have a few suggestions for the authors: - To give a more complete and effective picture of CVD risk in the studied population, I think that collection of CVD oriented family history and evaluation of total cholesterol could have been performed. I think that family history for CVD is one of the most important risk factors for CVD. Collecting a tailored family history is an economic and quick method to select families at high CVD risk. Moreover, I would suggest to test total cholesterol in the study participants through blood spot analysis, that can be performed quickly and during the same day of the cardiologic evaluation.- In table 1, I suggest that BMI of children should be better expressed as z-score, as BMI curves varies greatly according to age and sex. The same for systolic and diastolic blood pressure in children. Best regards, Maria Elena Capra
--

REVIEWER	DIPIKA BANSAL National Institute of Pharmaceutical Education and Research, Department of Pharmacy Practice
REVIEW RETURNED	02-Mar-2022

GENERAL COMMENTS	Page & Line No.	Comments
		Abstract
	Page 2 & Line No. 19	Kindly, simplify the sentence to make it meaningful.
	Introduction/Background	
	Page 3 & Line No. 21	Please reduce the use of punctuations, similarly in line 51 also (overuse of punctuations)
	Page 3& Line No. 27	Kindly explain more in account to “Heritability” in the scientific background.
	Methods	
	Page 4 & Line No. 49	Please mention the setting, locations more specifically whether its outpatient or inpatient units, government or private hospital etc.
	Page 5 & Line No. 23	Kindly mention the eligibility criteria, and the sources and methods of selection of participants clearly.
	Results:	
	Page 9 & Line No. 33	Kindly mention the number of male participants and their demographic details (include in table 1)
	Discussion	
	Page 12 & Line No. 18	Kindly provide the correct abbreviation (cIMT or CIMT)
	Page 13 & Line No. 3	Please discuss the generalisability (external validity) of the study results.

VERSION 1 – AUTHOR RESPONSE

1. Reviewer: 1 Dr. Maria Capra, Guglielmo da Saliceto Hospital

Comments to the Author:

The authors of this interesting article conduct a study to determine the heritability of phenotypic markers of cardiovascular risk across three generations. In their manuscript, they state that in South Africa “one in five deaths are from cardiovascular disease”, so cardiovascular disease (CVD) is an issue of utmost importance in this Country. I find this article interesting as it gives us a perspective of

cardiovascular risk in three generations, even if the generations are only female, as it is stated in the study limitations. I think this article is written accurately.

a. We thank the reviewer for these comments.

2. However, I have a few suggestions for the authors: To give a more complete and effective picture of CVD risk in the studied population, I think that collection of CVD oriented family history and evaluation of total cholesterol could have been performed. I think that family history for CVD is one of the most important risk factors for CVD. Collecting a tailored family history is an economic and quick method to select families at high CVD risk. Moreover, I would suggest to test total cholesterol in the study participants through blood spot analysis, that can be performed quickly and during the same day of the cardiologic evaluation.

a. Thank you for this suggestion. We agree this is a limitation and have added the following text to the limitation section within the discussion: "We also did not collect data on family history or blood markers of cardiovascular risk such as cholesterol within this study. Future studies should consider inclusion of a full CVD risk panel."

3. In table 1, I suggest that BMI of children should be better expressed as z-score, as BMI curves varies greatly according to age and sex. The same for systolic and diastolic blood pressure in children.

a. Thank you for this comment. We agree it is important to assess paediatric data according to age and sex. This has been done to categorize both BMI and blood pressure and is described in the methods with the following text: "For children, elevated blood pressure was defined using the age, sex, and height adjusted percentiles of the American Academy of Pediatrics Clinical Practice Guideline (2017)". "Children's BMI was categorised as underweight, normal, overweight or obese using age- and sex-specific cut-offs from the International Obesity Task Force (IOTF)". Good levels of agreement have been found between the IOTF cut-offs for BMI and the WHO z-scores in South African children [Jinabhai et al, Eur J Clin Nutr, 2003].

4. Reviewer: 2 - Dr. DIPIKA BANSAL, National Institute of Pharmaceutical Education and Research

Comments to the Author: After reviewing this manuscript, it seems that write up is up to the mark. The introduction is relevant. Overall, this manuscript can be considered for publication after minor modifications.

a. We thank the reviewer for these comments.

5. Abstract Page 2 & Line No. 19 Kindly, simplify the sentence to make it meaningful.

a. Thank you, while we are unable to see the same line numbers, we assume it is the sentence below that is problematic – "Heritability estimates were 23-44% for cIMT and 21-39% for LVMI." We have revised this to provide more context: "Heritability, the proportion of phenotypic trait variation attributable to genetics, was estimated from three common statistical methods and ranged from 23 to 44% for cIMT and from 21 to 39% for LVMI."

6. Introduction/Background Page 3 & Line No. 21 Please reduce the use of punctuations, similarly in line 51 also (overuse of punctuations)

a. We have tried to identify the section referred to and have changed the sentence with multiple punctuation to several shorter sentences, now reading: "For example, central blood pressures may show stronger heritability than the brachial blood pressures typically measured in routine care.¹⁶ Also carotid artery structure, function and pathology have been shown as heritable, with diameter and carotid intima media thickness appearing as the most heritable traits.¹⁷⁻¹⁹ Furthermore, arterial stiffness, as assessed by pulse wave velocity, has also been reported as heritable within family studies^{16,20} and findings from echocardiography studies suggest that several cardiac measurement

parameters may be heritable within families, including left ventricular (LV) function and structure including LV mass and LV hypertrophy²¹⁻²⁴.” Please also refer to point 7 above.

7. Page 3 & Line No. 27 Kindly explain more in account to “Heritability” in the scientific background.

a. Thank you for highlighting this gap. We have added the following text in the introduction: “Estimation of heritability or the proportion of variation in a phenotypic trait between individuals that is attributable to genetic factors, has been used for many years to predict disease risk in medicine⁵. While there may be debate regarding the exact measurement of genetic, environmental and interaction effects on trait variability, broadly heritability indicates the degree of resemblance of a trait within biological families⁶.”

8. Methods Page 4 & Line No. 49 Please mention the setting, locations more specifically whether its outpatient or inpatient units, government or private hospital etc.

a. The following detail has been added: “a research unit located within the grounds but operating independently of the outpatient and inpatient services of a large tertiary government hospital in Soweto, a historically disadvantaged township in South Africa.”

9. Page 5 & Line No. 23 Kindly mention the eligibility criteria, and the sources and methods of selection of participants clearly.

a. We have attempted to clarify this section which now reads: “...the Birth to Twenty Plus (Bt20-plus) cohort described in detail previously^{30 31}. Families in this cohort are tracked over time through engagement in ongoing assessments. In 2019, a database of 162 Bt20-plus index children (now the mothers) was drawn from all previous Birth to Twenty assessments that indicated both survival of their biological mother and birth of a biological child. These index children were then contacted by telephone to confirm the presence of their biological mother, and a biological child between the ages of 4 and 9 years. Families with participants who were pregnant, experiencing current acute illness, or with any major congenital disorders were excluded. All eligible families were invited to take part.”

10. Results: Page 9 & Line No. 33 Kindly mention the number of male participants and their demographic details (include in table 1)

a. Thank you for this comment. Sex is listed as a variable in Table 1 and to clarify this further we have added the following text to the Results: “All parents and grandparents were female, while 45% of children were male.” This lack of including fathers and grandfathers is discussed as a limitation within this study, though is a common research challenge in South Africa where statistics report up to 70% of black children do not live with their biological father [STATSSA 2021].

11. Discussion Page 12 & Line No. 18 Kindly provide the correct abbreviation (cIMT or CIMT)

a. Thank you , this has been corrected.

12. Page 13 & Line No. 3 Please discuss the generalisability (external validity) of the study results.

a. The following text has been added to the discussion showing the limitation in generalisability while more specifically stating our study population for reference: “Furthermore, it is noted that heritability estimates between and within populations are not constant and are influenced by factors such as environmental changes and migration⁵. While this may limit the generalisability of findings from any one study, it remains that heritability estimates for these cardiovascular phenotypes appear largely similar across many of the studies, regions, and populations. A key strength of this research is the contribution of evidence for the heritability and intergenerational transmission of cardiovascular health in black African families living in an urban African township”

VERSION 2 – REVIEW

REVIEWER	Maria Capra Guglielmo da Saliceto Hospital
REVIEW RETURNED	24-Aug-2022
GENERAL COMMENTS	Thank you for modifying your article according to my comments, I am satisfied with the revised version of the article.